# Sampled Estimators For Softmax Must Be Biased

**Li-Chung Lin**
National Taiwan University
r08922141@ntu.edu.tw

**Yaxu Liu**[*]
National Taiwan University
d08944045@ntu.edu.tw
Mohamed bin Zayed University of Artificial Intelligence
yaxu.liu@mbzuai.ac.ae

**Chih-Jen Lin**
National Taiwan University
cjlin@csie.ntu.edu.tw
Mohamed bin Zayed University of Artificial Intelligence
chihjen.lin@mbzuai.ac.ae

## Abstract

Models requiring probabilistic outputs are ubiquitous and used in fields such as natural language processing, contrastive learning, and recommendation systems. The standard method of designing such a model is to output unconstrained logits, which are normalized into probabilities with the softmax function. The normalization involves computing a summation across all classes, which becomes prohibitively expensive for problems with a large number of classes. An important strategy to reduce the cost is to sum over a sampled subset of classes in the softmax function, known as the sampled softmax. It was known that the sampled softmax is biased; the expectation taken over the sampled classes is not equal to the softmax function. Many works focused on reducing the bias by using a better way of sampling the subset. However, while sampled softmax is biased, it is unclear whether an unbiased function different from sampled softmax exists. In this paper, we show that all functions that only access a sampled subset of classes must be biased. With this result, we prevent efforts in finding unbiased loss functions and validate that past efforts devoted to reducing bias are the best we can do.

## 1 Introduction

Training losses based on the softmax function are extensively used across various fields, e.g., natural language processing [12, 13, 17, 1], contrastive learning [5, 4], and recommendation systems [6, 22]. The most common loss is to take the log of the softmax function, known as the log-softmax or cross-entropy loss,

$$L(\hat{y}^+, \hat{y}_1^-, \ldots, \hat{y}_n^-) = \log \text{Softmax}(\hat{y}^+, \hat{y}_1^-, \ldots, \hat{y}_n^-).$$

In this loss, the softmax function gives the probability of a class by

$$\text{Softmax}(\hat{y}^+, \hat{y}_1^-, \ldots, \hat{y}_n^-) = \frac{e^{\hat{y}^+}}{e^{\hat{y}^+} + \sum_{i=1}^n e^{\hat{y}_i^-}}, \tag{1}$$

where $\hat{y}^+ \in \mathbb{R}$ is the logit of the target class, $\hat{y}_i^- \in \mathbb{R}$ for every $1 \leq i \leq n$ are the logits of all negative classes, and there are $n + 1$ total classes. By summing this loss over all training data, we obtain the overall learning objective.

---

[*]Work mainly done while visiting Mohamed bin Zayed University of Artificial Intelligence.

39th Conference on Neural Information Processing Systems (NeurIPS 2025).

One typically employs gradient methods to train machine learning models. At each training step, the gradient of log-softmax must be calculated, leading to the following partial derivatives[2]

$$\frac{\partial}{\partial \hat{y}^+} \log \text{Softmax}(\hat{y}^+, \hat{y}_1^-, \ldots, \hat{y}_n^-) = 1 - \text{Softmax}(\hat{y}^+, \hat{y}_1^-, \ldots, \hat{y}_n^-) \tag{2}$$

and for every $1 \leq i \leq n$,

$$\frac{\partial}{\partial \hat{y}_i^-} \log \text{Softmax}(\hat{y}^+, \hat{y}_1^-, \ldots, \hat{y}_n^-) = -\text{Softmax}(\hat{y}_i^-, \hat{y}_1^-, \ldots, \hat{y}_{i-1}^-, \hat{y}^+, \hat{y}_{i+1}^-, \ldots \hat{y}_n^-). \tag{3}$$

Both the loss and gradient computations inevitably rely on the softmax function. The denominator in (1) sums over logits for all $n$ negative classes, leading to a computational cost of $\mathcal{O}(n)$ for each calculation. This cost can become prohibitively expensive when training models on datasets with a large number of classes. A typical example involves recommendation systems, like [22], in which the size of $n$ can reach millions. In addition to the $\mathcal{O}(n)$ computational complexity of the denominator, evaluating (1) requires computing all $n$ logits, $\hat{y}_i^-$. When using large neural networks such as BERT [9], computing all $n$ logits can become prohibitively expensive.

As shown in [22], an intuitive idea to reduce the computation cost is to sample a subset of $k$ classes for loss calculation, and $k \ll n$ typically. An important scenario is the stochastic gradient descent method, where each training step involves a mini-batch of data, and within each batch only the sampled $k$ classes are used to compute the stochastic gradient. That is, we use only $k$ sampled classes in calculating (2) and (3), where a widely used setting is to consider the following sampled softmax:

$$\frac{e^{\hat{y}^+}}{e^{\hat{y}^+} + \sum_{j=1}^k w(i_j) e^{\hat{y}_{i_j}^-}}, \tag{4}$$

where $k$ classes $(i_1, \ldots, i_k)$ are randomly sampled from the $n$ classes and $w(i_j)$ is a weighting term inversely proportional to the probability of sampling the $i_j$th sampled class [2]. In contrast, (1) represents the *full softmax*, which involves all $n + 1$ classes.

To ensure equivalence with training a model using the full softmax, it is desirable for the sampled softmax to satisfy the condition of unbiasedness. Specifically, this condition is defined as follows:

$$\forall \hat{y}^+, \hat{y}_1^-, \ldots, \hat{y}_n^- \in \mathbb{R}, \ \mathbb{E}\left[\text{Stochastic Gradient}\right] = \text{Full Gradient}. \tag{5}$$

When applying (4), this condition implies that

$$\forall \hat{y}^+, \hat{y}_1^-, \ldots, \hat{y}_n^- \in \mathbb{R}, \ \mathbb{E}\left[(4)\right] = (1), \tag{6}$$

where $\mathbb{E}\left[\cdot\right]$ denotes the expectation over the sampled classes $(i_1, \ldots, i_k)$. If this condition does not hold, we consider that (4) is biased with respect to (1). The different gradients are potentially harmful to the optimization process and solutions. Unfortunately, previous works like [2, 3] have already demonstrated that (6) does not hold, highlighting the inherent bias of the sampled softmax. Moreover, experiments in [3, 16] have demonstrated that bias degrades model performance in tasks such as multi-label classification and content recommendation.

The bias of sampled softmax leads us to wonder whether, beyond (4), there could be other forms of approximation that are unbiased. In this paper, we show that any general function with access to a subset of logits must be biased. Specifically, when $k < n$, let

$$\mathbb{S}_k = \{s \mid s : \mathbb{R}^{k+1} \to \mathbb{R}\}$$

denote the set of all real-valued functions of $k + 1$ real inputs, which covers all possible estimators operating on one positive logit and $k$ sampled negative logits. Then,

$$\forall s \in \mathbb{S}_k, \ \exists \hat{y}^+, \hat{y}_1^-, \ldots, \hat{y}_n^- \in \mathbb{R}, \ \text{s.t. } \mathbb{E}\left[s(\hat{y}^+, \hat{y}_{i_1}^-, \ldots, \hat{y}_{i_k}^-)\right] \neq (1), \tag{7}$$

where $k$ classes $(i_1, \ldots, i_k)$ are randomly sampled from the $n$ classes. In other words, there is no estimator $s$ satisfying the unbiased condition for every possible set of $n$ logits when $s$ has access to only $k < n$ logits. By extending from the sampled softmax to any general function $s$, this conclusion generalizes the finding of [3] and forms our main contribution. Our negative answer to the existence

---
[2]The detailed proof is given in Appendix A.

Table 1: Notation

| Notation | Description |
|---|---|
| $n$ | Number of classes excluding the target |
| $\hat{y}^+ \in \mathbb{R}$ | Logit of the target class |
| $\hat{y}_1^-, \ldots, \hat{y}_n^- \in \mathbb{R}$ | Logits of the remaining classes |
| $k$ | Number of sampled classes |
| $[r] = \{1, \ldots, r\}$ | Set of positive integers up to $r$ |
| $S_r$ | Set of permutations on $[r]$ |
| $\binom{q}{r} = \dfrac{q!}{r!(q-r)!}$ | Number of combinations of $r$ items chosen from $q$ items |
| $\mathbb{R}_{>0} = \{a \in \mathbb{R} \mid a > 0\}$ | Set of positive real numbers |
| $h_k : \mathbb{R}^k \to \mathbb{R}$ | Estimator taking $k$ arguments |
| $f_k : \mathbb{R}^k \to \mathbb{R}$ | The symmetrization of $h_k$ |

of unbiased approximations establishes a theoretical boundary on what is and is not achievable. Our result not only fills a notable gap in the literature but also sets clear limits for future research directions.

The remainder of this paper is organized as follows: In Section 2, we summarize related works. In Section 3, we formally define the problem we aim to address and then state the main results of this work and our assumption. Section 4 starts with preliminaries in Section 4.1 that provide the necessary background and lemmas, followed by a sketch of the proof in Section 4.2. The detailed proof is presented in Sections 4.3 and Appendix E. Finally, Section 5 concludes the paper. The main notations used in this work are in Table 1, and the appendix is available at `https://www.csie.ntu.edu.tw/~cjlin/papers/softmax_biased/`.

## 2 Related Works

Prior efforts all implicitly assume that an unbiased solution does not exist without attempting to justify or challenge it. They can be roughly categorized into three directions.

- **Focusing on the sampled softmax (4) and attempting to minimize its bias.** While [3] demonstrated that the inherent bias in the sampled softmax cannot be completely eliminated, they proposed a strategy to minimize this bias. Specifically, the closer the sampling distribution used for selecting $(i_1, \ldots, i_k)$ approximates the full softmax, the less biased the sampled softmax becomes. To this end, [3] and its follow-up study [16] developed computationally efficient approximations for sampling. Additionally, numerous hard negative mining strategies have been widely adopted alongside the sampled softmax, such as those used in representation learning [21, 20, 7]. Hard negatives refer to negative classes that produce large gradients when used in the sampled log-softmax loss:

$$-\log\left(\frac{e^{\hat{y}^+}}{e^{\hat{y}^+} + \sum_{j=1}^{k} e^{\hat{y}_{i_j}^-}}\right).$$

  This setting effectively involves sampling negatives based on their corresponding logits. Furthermore, studies like [24] have argued that an effective hard negative mining method should be based on a good approximation of the full softmax distribution, consistent with the ideas proposed in [3].

  In addition to the aforementioned approach of selecting an appropriate sampling distribution, [3] mentioned another idea: using a larger sample size $k$ to reduce bias. However, increasing $k$ naturally leads to higher computational cost. To address this issue, subsequent works [19, 23] introduced the concept of applying moving averages over multiple mini-batches of gradients, thereby approximating the effect of a large sample size $k$ while maintaining low cost.

- **Exploring alternative functions that are computationally more efficient than the full softmax (1).** Works in this category generally try to avoid sampling and explore a new loss with all $n$ logits in their calculation. Well-known examples include hierarchical softmax [14] and spherical softmax [8]. However, as reported in [8], such alternative functions may perform worse than the

full softmax (1). Since these methods do not involve sampling over the $n$ classes, we do not extend our discussion to them.

- **Designing a different learning problem with the same optimal solution.** In [11, 15, 10], they transform the original optimization problem involving a log-softmax term in the objective function into a new optimization problem. In the transformed problem, there is no log-softmax term in the objective function. They then show that for special types of models, the transformed problem has the same optimal solution as the original problem. The equivalence of the optimal solution is what they refer to as "unbiased," which is unrelated to the problem of bias discussed in this paper. While these reformulation approaches are still under development, our negative answer to directly sampling softmax indicates that they are directions worth investigation.

By proving the non-existence of an unbiased solution, our work provides a theoretical boundary and serves as a meaningful complement to these studies.

## 3   Problem Definition and Main Result

Let $(i_1, \ldots, i_k) \in [n]^k$ be random variables denoting a subset of $k$ indices sampled from $[n] = \{1, \ldots, n\}$ without replacement. We wish to find an estimator $s(\hat{y}^+, \hat{y}_{i_1}^-, \ldots, \hat{y}_{i_k}^-)$ such that

$$\mathbb{E}\left[s(\hat{y}^+, \hat{y}_{i_1}^-, \ldots, \hat{y}_{i_k}^-)\right] = \frac{e^{\hat{y}^+}}{e^{\hat{y}^+} + \sum_{i=1}^n e^{\hat{y}_i^-}}, \tag{8}$$

where the expectation is over $(i_1, \ldots, i_k)$.[3]

Our main results are Theorem 3.1 and Corollary 3.3.

**Theorem 3.1.** *Let $k < n$ and $\mathbb{S}_k = \{s \mid s : \mathbb{R}^{k+1} \to \mathbb{R}\}$ denote the set of all real-valued functions of $k + 1$ real inputs. Consider the setting where $(i_1, \ldots, i_k)$ are sampled uniformly without replacement from $\{1, \ldots, n\}$, and an estimator $s \in \mathcal{S}_k$ is applied to one positive logit $\hat{y}^+$ and the $k$ sampled negative logits $(\hat{y}_{i_1}^-, \ldots, \hat{y}_{i_k}^-)$. Then*

$$\forall s \in \mathbb{S}_k, \ \exists \hat{y}^+, \hat{y}_1^-, \ldots, \hat{y}_n^- \in \mathbb{R}, \ s.t. \ \mathbb{E}\left[s(\hat{y}^+, \hat{y}_{i_1}^-, \ldots, \hat{y}_{i_k}^-)\right] \neq \text{Softmax}(\hat{y}^+, \hat{y}_1^-, \ldots, \hat{y}_n^-), \tag{9}$$

*where $\mathbb{E}[\cdot]$ is taken over the random choice of $(i_1, \ldots, i_k)$.*

In this work, we focus on the existence of an unbiased estimator of the softmax function for two main reasons. First, during training, what matters most is the gradient of log-softmax, as model parameters are updated through gradient-based optimization. Second, during inference, the output of interest is softmax itself, which represents the predicted probability distribution over classes. In both situations, the exact value of the log-softmax is not directly required. Regardless, with Theorem 3.1, we show that an unbiased estimator of log-softmax does not exist in Appendix F.

After we establish that Theorem 3.1 applies for uniform sampling, we next examine whether Theorem 3.1 still holds for other ways of sampling the $k$ classes $(i_1, \ldots, i_k)$. An example of non-uniform sampling is [13], where the samples are the union of the target class, in-batch negative samples, and samples based on BM25 [18]. In-batch negative samples refer to using the target classes of other instances in a mini-batch as the negative classes of the current instance.

While we want to consider sampling distributions that are as general as possible, we must be careful not to include impractical sampling distributions. Recall that the purpose of a sampled estimator is to avoid computing all $n$ logits; using the logits as part of the sampling process defeats the entire point of sampling. Consequently, we make the following assumption.

**Assumption 3.2.** The sampling distribution is selected before computing the logits.

An example of violating Assumption 3.2 is to use a distribution where the probability of sampling each class $i$ is proportional to the exponential of its logit $e^{y_i}$. In such a case, it was proven [3] that the sampled softmax is an unbiased estimator. However, constructing the distribution is equivalent to computing the softmax function for each class. Clearly, if we already have the softmax function, we

---

[3]Note that (4) with $w(i_j)$ can be also expressed by (8). Details are in Appendix B.

would not need to compute a sampled estimator. We obviously want to exclude such distributions from consideration.

In the previous example of [13], in-batch sampling and negative sampling are both selected before computing the logits, so they satisfy Assumption 3.2. Similarly, we find that most works on sampled softmax satisfy Assumption 3.2. For the minority of exceptions [3, 16], we give a brief discussion in Appendix D.

Given Assumption 3.2 and setting $s(\cdot)$ to (4), [3] concluded that the sampled softmax is biased with respect to the full softmax,

$$\mathbb{E}\left[\frac{e^{\hat{y}^+}}{e^{\hat{y}^+} + \sum_{j=1}^{k} w(i_j)e^{\hat{y}_{i_j}}}\right] \neq \frac{e^{\hat{y}^+}}{e^{\hat{y}^+} + \sum_{i=1}^{n} e^{\hat{y}_i}}.$$

In the following sections, we further extend their conclusion to any general $s(\cdot)$ under Assumption 3.2.

**Corollary 3.3.** *Theorem 3.1 also holds if the indices $(i_1, \ldots, i_k)$ are sampled from any distribution satisfying Assumption 3.2.*

The proof of Corollary 3.3 is given in Appendix E.

In the following proof of (7), due to a simplification outlined in Section 4.2, we will work with a function $h_k : \mathbb{R}^k \to \mathbb{R}$ instead of $s : \mathbb{R}^{k+1} \to \mathbb{R}$, but they refer to the same problem of estimating the full softmax.

## 4 Main Proof

In this section, we prove Theorem 3.1. We begin with necessary preliminaries and lemmas, followed by a sketch that highlights the main ideas of the proof before presenting the complete derivation.

### 4.1 Preliminaries

This section introduces lemmas used in the proof of the main theorem. The proofs are provided in Appendix C.

**Definition 4.1.** Let $h_k : \mathbb{R}^k \to \mathbb{R}$ be a function and $S_k$ be the set of permutations on $[k]$. The symmetrization $f_k$ of $h_k$ is defined by

$$f_k(x_1, \ldots, x_k) \equiv \frac{1}{k!} \sum_{\tau \in S_k} h_k(x_{\tau(1)}, \ldots, x_{\tau(k)}).$$

The symmetrization is so named because it is symmetric in its arguments, stated as follows.

**Lemma 4.2.** *For every $\sigma \in S_k$,*

$$f_k(x_1, \ldots, x_k) = f_k(x_{\sigma(1)}, \ldots, x_{\sigma(k)}).$$

Furthermore, the symmetrization has the same expectation under a uniform sample, stated as follows.

**Lemma 4.3.** *Let $\mathbf{a} = (a_1, \ldots, a_n) \in \mathbb{R}^n$ be $n$ real numbers and $(i_1, \ldots, i_k) \in [n]^k$ be random variables denoting $k$ indices sampled uniformly from $[n]$ without replacement. Then*

$$\mathbb{E}\left[f_k(a_{i_1}, \ldots, a_{i_k})\right] = \mathbb{E}\left[h_k(a_{i_1}, \ldots, a_{i_k})\right],$$

*where the expectation is over $(i_1, \ldots, i_k)$.*

### 4.2 Sketch of Proof for Theorem 3.1

The core idea of our proof is that unbiasedness imposes a strong requirement, demanding the estimator to generalize across arbitrary cases. To illustrate this, we constructed two of the simplest yet representative cases: one where all logits are identical and another where the logits are divided

into two groups, each with a distinct value. It turns out that these two minimal cases alone have already revealed a set of contradictory constraints. The proof is non-trivial, as it involves identifying subtle examples that clearly demonstrate the inherent contradictions.

The proof of Theorem 3.1 proceeds by contradiction, as shown below:

1. We begin by reformulating the problem of estimating the full softmax into an equivalent problem of estimating:

$$F(\boldsymbol{a}) = \frac{1}{1 + \sum_{i=1}^n a_i},$$

   where $\boldsymbol{a} = (a_1, \ldots, a_n) \in \mathbb{R}_{>0}^n$ and $\mathbb{R}_{>0} = \{a \in \mathbb{R} \mid a > 0\}$.

2. We hypothesize the existence of an unbiased estimator $h_k : \mathbb{R}^k \to \mathbb{R}$ for $F$, which accepts $k$ arguments, and uses the symmetrization $f_k$ of $h_k$, which has the same expectation, to simplify the derivations.

3. We analyze the expectation of $f_k$ across various possible values of $\boldsymbol{a}$ and derive a necessary equation for each vector $\boldsymbol{a}$ that $f_k$ must satisfy to be unbiased. These equations are derived using mathematical induction.

4. Finally, we complete our proof by demonstrating that it is impossible for $f_k$ to simultaneously satisfy all equations, leading to a contradiction.

Specifically, for Step 3 above, we check different $\boldsymbol{a}$ of the form,

$$(b_1, \ldots, b_1, \underbrace{b_2, \ldots, b_2}_{m}),$$

where $b_1, b_2 \in \mathbb{R}_{>0}$ and $m = 0, \ldots, k$.

### 4.3 Proof of Theorem 3.1

*Proof.* First, we note that

$$\text{Softmax}(\hat{y}^+, \hat{y}_1^-, \ldots, \hat{y}_n^-) = \frac{e^{\hat{y}^+}}{e^{\hat{y}^+} + \sum_{i=1}^n e^{\hat{y}_i^-}} = \frac{1}{1 + \sum_{i=1}^n e^{\hat{y}_i^- - \hat{y}^+}}.$$

It follows that there is an unbiased estimator for the full softmax for every possible model output if and only if there is an unbiased estimator for $F$

$$F(\boldsymbol{a}) = \frac{1}{1 + \sum_{i=1}^n a_i} \tag{10}$$

for every $\boldsymbol{a} = (a_1, \ldots, a_n) \in \mathbb{R}_{>0}^n$.

We prove by contradiction, so we assume there is an unbiased estimator $h_k$. That is, there exists an $h_k$ such that for every $\boldsymbol{a}$

$$F(\boldsymbol{a}) = \mathbb{E}\left[h_k(a_{i_1}, \ldots, a_{i_k})\right], \tag{11}$$

where $(i_1, \ldots, i_k) \in [n]^k$ are random variables denoting $k$ indices sampled uniformly from $[n]$ without replacement.

Next, we calculate the expectation in (11) in detail. For the selection of $(i_1, \ldots, i_k)$, the number of all possible samples of indices is

$$\binom{n}{k} \cdot k! = \frac{n!}{(n-k)!},$$

which is the number of combinations of $k$ choices from $n$ items multiplied by the number of permutations on $k$ items. Therefore, for every function $G : \mathbb{R}^k \to \mathbb{R}$ we have

$$\mathbb{E}\left[G(a_{i_1}, \ldots, a_{i_k})\right] = \frac{(n-k)!}{n!} \sum_{\boldsymbol{x} \in C(n,k;\boldsymbol{a})} \sum_{\sigma \in S_k} G(x_{\sigma(1)}, \ldots, x_{\sigma(k)}), \tag{12}$$

where $C(n, k; \boldsymbol{a})$ denotes the set of $\binom{n}{k}$ choices of $k$ elements from $\boldsymbol{a}$.

Let $f_k$ be the symmetrization of $h_k$. Then

$$F(\boldsymbol{a}) = \mathbb{E}\left[h_k(a_{i_1}, \ldots, a_{i_k})\right] = \mathbb{E}\left[f_k(a_{i_1}, \ldots, a_{i_k})\right] \tag{13}$$

$$= \frac{(n-k)!}{n!} \sum_{\boldsymbol{x} \in C(n,k;\boldsymbol{a})} \sum_{\sigma \in S_k} f_k(x_{\sigma(1)}, \ldots, x_{\sigma(k)})$$

$$= \frac{(n-k)!}{n!} \sum_{\boldsymbol{x} \in C(n,k;\boldsymbol{a})} \sum_{\sigma \in S_k} f_k(x_1, \ldots, x_k) \tag{14}$$

$$= \frac{(n-k)!k!}{n!} \sum_{\boldsymbol{x} \in C(n,k;\boldsymbol{a})} f_k(x_1, \ldots, x_k)$$

$$= \frac{1}{\binom{n}{k}} \sum_{\boldsymbol{x} \in C(n,k;\boldsymbol{a})} f_k(x_1, \ldots, x_k), \tag{15}$$

where (13) follows from Lemma 4.3 and (14) follows from Lemma 4.2.

We claim that for every $b_1 \in \mathbb{R}_{>0}$, we have

$$f_k(b_1, \ldots, b_1) = \frac{1}{1 + nb_1}. \tag{16}$$

This is because $f_k$ is unbiased for every $\boldsymbol{a}$, including for $\boldsymbol{a} = (b_1, \ldots, b_1)$, implying

$$F(\boldsymbol{a}) = \frac{1}{1 + nb_1} \tag{17}$$

$$= \mathbb{E}\left[f_k(a_{i_1}, \ldots, a_{i_k})\right] \tag{18}$$

$$= \frac{1}{\binom{n}{k}} \sum_{\boldsymbol{x} \in C(n,k;\boldsymbol{a})} f_k(b_1, \ldots, b_1) \tag{19}$$

$$= f_k(b_1, \ldots, b_1),$$

where (17) is the definition of $F$ in (10), (18) is from (13) and (19) is from (15).

Similarly, suppose $\boldsymbol{a} = (b_1, \ldots, b_1, b_2)$ for some $b_1, b_2 \in \mathbb{R}_{>0}$. Then we have

$$F(\boldsymbol{a}) = \frac{1}{1 + (n-1)b_1 + b_2}$$

$$= \mathbb{E}\left[f_k(a_{i_1}, \ldots, a_{i_k})\right]$$

$$= \frac{1}{\binom{n}{k}} \left( \binom{n-1}{k} f_k(b_1, \ldots, b_1) + \binom{n-1}{k-1} f_k(b_1, \ldots, b_1, b_2) \right) \tag{20}$$

$$= \frac{(n-k)!k!}{n!} \frac{(n-1)!}{(n-k-1)!k!} \frac{1}{1 + nb_1} + \frac{(n-k)!k!}{n!} \frac{(n-1)!}{(n-k)!(k-1)!} f_k(b_1, \ldots, b_1, b_2) \tag{21}$$

$$= \frac{n-k}{n} \frac{1}{1 + nb_1} + \frac{k}{n} f_k(b_1, \ldots, b_1, b_2).$$

The term $\binom{n-1}{k}$ in (20) is the number of ways of choosing $k$ counts of $b_1$ and the term $\binom{n-1}{k-1}$ is the number of ways of choosing $k-1$ counts of $b_1$ and 1 count of $b_2$. Also, in (21), we use (16). Rearranging, we have

$$f_k(b_1, \ldots, b_1, b_2) = \frac{n}{k} \frac{1}{1 + (n-1)b_1 + b_2} - \frac{n-k}{k} \frac{1}{1 + nb_1}.$$

More generally, for each $m \leq k$, if we consider

$$\boldsymbol{a} = (b_1, \ldots, b_1, \underbrace{b_2, \ldots, b_2}_{m}),$$

we find that

$$F(\boldsymbol{a}) = \frac{1}{1 + (n-m)b_1 + mb_2} = \mathbb{E}\left[f_k(a_{i_1}, \ldots, a_{i_k})\right] = \frac{1}{\binom{n}{k}} \sum_{j=0}^{m} \binom{n-m}{k-j}\binom{m}{j} g(j), \quad (22)$$

where

$$g(j) \equiv f_k(b_1, \ldots, b_1, \underbrace{b_2, \ldots, b_2}_{j}) \qquad j = 0, \ldots, k, \quad (23)$$

and the factor $\binom{n-m}{k-j}\binom{m}{j}$ is the number of ways of choosing $k - j$ counts of $b_1$ and $j$ counts of $b_2$. The dependence of $g$ on $n$, $k$, $b_1$ and $b_2$ are suppressed in the notation for brevity. We prove by strong induction on $m$ that

$$g(m) = \sum_{l=0}^{m} \frac{c(m,l)}{1 + (n-l)b_1 + lb_2} \qquad m = 0, \ldots, k, \quad (24)$$

where

$$c(m,l) \quad (25)$$

$$= \begin{cases} \dfrac{\binom{n}{k}}{\binom{n-m}{k-m}} & \text{if } m = l, \\[3ex] -\dfrac{1}{\binom{n-m}{k-m}} \displaystyle\sum_{j=l}^{m-1} \binom{n-m}{k-j}\binom{m}{j} c(j,l) & \text{if } m > l, \\[3ex] 0 & \text{otherwise.} \end{cases}$$

Likewise, the dependence of $c$ on $n$ and $k$ are suppressed in the notation for brevity.

For the base case,

$$g(0) = f_k(b_1, \ldots, b_1) = \frac{1}{1 + nb_1} = \frac{c(0,0)}{1 + nb_1},$$

where we respectively use (23), (16) and

$$c(0,0) = \binom{n}{k} \Big/ \binom{n}{k} = 1 \quad (26)$$

for the derivation above.

For the induction step, we split the summation in (22) to single out $g(m)$:

$$\frac{1}{\binom{n}{k}} \binom{n-m}{k-m}\binom{m}{m} g(m) + \frac{1}{\binom{n}{k}} \sum_{j=0}^{m-1} \binom{n-m}{k-j}\binom{m}{j} g(j). \quad (27)$$

Rearranging (22), we have

$$g(m) = \frac{\binom{n}{k}}{\binom{n-m}{k-m}\binom{m}{m}} \left( \frac{1}{1 + (n-m)b_1 + mb_2} - \frac{1}{\binom{n}{k}} \sum_{j=0}^{m-1} \binom{n-m}{k-j}\binom{m}{j} g(j) \right)$$

$$= \frac{\binom{n}{k}}{\binom{n-m}{k-m}} \frac{1}{1 + (n-m)b_1 + mb_2} - \frac{1}{\binom{n-m}{k-m}} \sum_{j=0}^{m-1} \binom{n-m}{k-j}\binom{m}{j} g(j). \quad (28)$$

For the first term in (28), by the definition in (25), we have

$$c(m,m) = \frac{\binom{n}{k}}{\binom{n-m}{k-m}}. \quad (29)$$

For the second term, assuming the induction hypothesis (24) is true for all $j = 0, \ldots, m-1$, we have

$$- \frac{1}{\binom{n-m}{k-m}} \sum_{j=0}^{m-1} \binom{n-m}{k-j}\binom{m}{j} g(j)$$

$$= \frac{-1}{\binom{n-m}{k-m}} \sum_{j=0}^{m-1} \binom{n-m}{k-j}\binom{m}{j} \sum_{l=0}^{j} \frac{c(j,l)}{1 + (n-l)b_1 + lb_2} \tag{30}$$

$$= \frac{-1}{\binom{n-m}{k-m}} \sum_{j=0}^{m-1} \sum_{l=0}^{j} \binom{n-m}{k-j}\binom{m}{j} \frac{c(j,l)}{1 + (n-l)b_1 + lb_2}$$

$$= \frac{-1}{\binom{n-m}{k-m}} \sum_{l=0}^{m-1} \sum_{j=l}^{m-1} \binom{n-m}{k-j}\binom{m}{j} \frac{c(j,l)}{1 + (n-l)b_1 + lb_2} \tag{31}$$

$$= \sum_{l=0}^{m-1} \frac{\frac{-1}{\binom{n-m}{k-m}} \sum_{j=l}^{m-1} \binom{n-m}{k-j}\binom{m}{j} c(j,l)}{1 + (n-l)b_1 + lb_2}$$

$$= \sum_{l=0}^{m-1} \frac{c(m,l)}{1 + (n-l)b_1 + lb_2}. \tag{32}$$

In (30), $g(j)$ is substituted with the induction hypothesis (24); (31) exchanges the order of summation over $j$ and $l$; and $c(m,l)$ in (32) follows from the definition in (25). By replacing values in (28) with (29) and (32), we have

$$g(m) = \frac{\binom{n}{k}}{\binom{n-m}{k-m}} \frac{1}{1 + (n-m)b_1 + mb_2} - \frac{1}{\binom{n-m}{k-m}} \sum_{j=0}^{m-1} \binom{n-m}{k-j}\binom{m}{j} g(j)$$

$$= \frac{c(m,m)}{1 + (n-m)b_1 + mb_2} + \sum_{l=0}^{m-1} \frac{c(m,l)}{1 + (n-l)b_1 + lb_2}$$

$$= \sum_{l=0}^{m} \frac{c(m,l)}{1 + (n-l)b_1 + lb_2}.$$

Therefore, we have finished proving (24) by induction.

Let

$$c^* = \max_{j,l \le k} |c(j,l)|.$$

Recalling that $k < n$, we have for all $j \le k$

$$|g(j)| = \left| \sum_{l=0}^{j} \frac{c(j,l)}{1 + (n-l)b_1 + lb_2} \right|$$

$$\le \sum_{l=0}^{j} \frac{|c(j,l)|}{1 + (n-l)b_1 + lb_2} \tag{33}$$

$$\le \sum_{l=0}^{j} \frac{c^*}{1 + (n-l)b_1 + lb_2}$$

$$< \sum_{l=0}^{j} \frac{c^*}{1 + (n-k)b_1} \tag{34}$$

$$= \frac{(j+1)c^*}{1 + (n-k)b_1} \qquad j = 0, \ldots, k, \tag{35}$$

where (33) is due to triangle inequality, and (34) follows from the facts that $b_1, b_2 \in \mathbb{R}_{>0}$ and $k > 0$, since $k$ is the number of samples. However, by the definition of $g$ in (23) and (16),

$$g(k) = f_k(b_2, \ldots, b_2) = \frac{1}{1 + nb_2}.$$

So we can use (35) to have

$$\frac{1}{1 + nb_2} = |g(k)| < \frac{(k+1)c^*}{1 + (n-k)b_1},$$

which is a contradiction because if

$$b_1 = \frac{2(k+1)c^* - 1}{n - k}$$

and

$$b_2 = \frac{1}{n},$$

then

$$\frac{1}{1 + nb_2} = \frac{1}{2} = \frac{(k+1)c^*}{1 + (n-k)b_1}.$$

To show that $b_1$ is well-defined, i.e., $b_1 > 0$, we have

$$
\begin{aligned}
b_1 &= \frac{2(k+1)c^* - 1}{n - k} \\
&= \frac{(2(k+1)\max_{j,l \le k} |c(j,l)|) - 1}{n - k} \\
&\ge \frac{2(k+1)|c(0,0)| - 1}{n - k} \\
&= \frac{2(k+1) - 1}{n - k} \\
&> 0,
\end{aligned}
\tag{36}
$$

where (36) follows from (26). The contradiction completes our proof. □

## 5  Conclusions

This paper considered the problem of bias for methods that sample classes and examined the possibility of using a different function instead of the sampled softmax. Previous works have taken it as a foundational assumption that such a function cannot be found, treating this claim as if it were true without justification. We proved, for the first time, that no function can be an unbiased estimator, which extends the conclusion in [3] that is only limited to the sampled softmax. Our result now can serve as the theoretical premise for related studies. According to our result, future work should not aim to find an unbiased estimator. By providing a proof of impossibility, our work allows follow-up studies to focus on feasible solutions.

## Acknowledgements

This work was supported in part by National Science and Technology Council of Taiwan grant NSTC-113-2222-E-002- 005-MY3, and in part by the Featured Area Research Center Program within the framework of the Higher Education Sprout Project by the Ministry of Education (114L900901). The authors thank the reviewers for their insightful and constructive comments.

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

# A  Using Log-Softmax Loss During Training

The typical loss function for models based on Softmax is the log-softmax function

$$L(\hat{y}^+, \hat{y}_1^-, \ldots, \hat{y}_n^-) = \log \mathrm{Softmax}(\hat{y}^+, \hat{y}_1^-, \ldots, \hat{y}_n^-)$$

$$= \hat{y}^+ - \log\left(e^{\hat{y}^+} + \sum_{i=1}^n e^{\hat{y}_i^-}\right).$$

During training, we need to compute the gradient of $L$ at each step. In other words, we need to compute the partial derivatives. The partial derivative of log-softmax with respect to the target class is

$$\frac{\partial L(\hat{y}^+, \hat{y}_1^-, \ldots, \hat{y}_n^-)}{\partial \hat{y}^+} = 1 - \frac{e^{\hat{y}^+}}{e^{\hat{y}^+} + \sum_{i=1}^n e^{\hat{y}_i^-}}$$

$$= 1 - \mathrm{Softmax}(\hat{y}^+, \hat{y}_1^-, \ldots, \hat{y}_n^-),$$

and the partial derivatives with respect to the negative classes $j \in \{1, \ldots, n\}$ are

$$\frac{\partial L(\hat{y}^+, \hat{y}_1^-, \ldots, \hat{y}_n^-)}{\partial \hat{y}_j^-} = -\frac{e^{\hat{y}_j^-}}{e^{\hat{y}^+} + \sum_{i=1}^n e^{\hat{y}_i^-}}$$

$$= -\mathrm{Softmax}(\hat{y}_j^-, \hat{y}_1^-, \ldots, \hat{y}_{j-1}^-, \hat{y}^+, \hat{y}_{j+1}^-, \ldots \hat{y}_n^-).$$

Therefore, computing the gradient of the loss function requires computing Softmax.

# B  Reducing (4) to (8)

Let $\hat{o}_i^- \in \mathbb{R}$ denote the original logits for the $i$th negative class. Instead of using $\hat{o}_i^-$ directly, we use

$$\hat{y}_i^- = \hat{o}_i^- + \log(w(i))$$

such that

$$e^{\hat{y}_i^-} = e^{\hat{o}_i^- + \log(w(i))}$$

$$= w(i)\, e^{\hat{o}_i^-},$$

where $\forall i, w(i)$ are constants. So we have

$$s(\hat{y}^+, \hat{y}_{i_1}^-, \ldots, \hat{y}_{i_k}^-) = \frac{e^{\hat{y}^+}}{e^{\hat{y}^+} + \sum_{j=1}^k e^{\hat{y}_{i_j}^-}}$$

$$= \frac{e^{\hat{y}^+}}{e^{\hat{y}^+} + \sum_{j=1}^k w(i_j) e^{\hat{o}_{i_j}^-}},$$

which is the sampled softmax in (4) on the original logits $\hat{o}_i^-$.

# C  Proof of Lemmas

## C.1  Proof of Lemma 4.2

*Proof.*

$$f_k(x_1, \ldots, x_k) = \frac{1}{k!} \sum_{\tau \in S_k} h_k(x_{\tau(1)}, \ldots, x_{\tau(k)})$$

$$= \frac{1}{k!} \sum_{\tau \in S_k} h_k(x_{\tau(\sigma(1))}, \ldots, x_{\tau(\sigma(k))}) \tag{37}$$

$$= f_k(x_{\sigma(1)}, \ldots, x_{\sigma(k)}),$$

where (37) follows because the summation over $\tau$ sums over the same set of summands.  □

## C.2 Proof of Lemma 4.3

*Proof.* We first note that the number of all possible samples of indices is

$$\binom{n}{k} \cdot k! = \frac{n!}{(n-k)!},$$

which is the number of combinations of $k$ choices from $n$ items multiplied by the number of permutations on $k$ items. Let $C(n, k; \boldsymbol{a})$ denote the set of $\binom{n}{k}$ choices of $k$ elements from $\boldsymbol{a}$. Then

$$
\begin{aligned}
\mathbb{E}\left[f_k(a_{i_1}, \ldots, a_{i_k})\right] &= \frac{(n-k)!}{n!} \sum_{\boldsymbol{x} \in C(n,k;\boldsymbol{a})} \sum_{\sigma \in S_k} f_k(x_{\sigma(1)}, \ldots, x_{\sigma(k)}) \\
&= \frac{(n-k)!}{n!} \sum_{\boldsymbol{x} \in C(n,k;\boldsymbol{a})} \sum_{\sigma \in S_k} \frac{1}{k!} \sum_{\tau \in S_k} h_k(x_{\tau(\sigma(1))}, \ldots, x_{\tau(\sigma(k))}) \\
&= \frac{(n-k)!}{n!} \sum_{\boldsymbol{x} \in C(n,k;\boldsymbol{a})} \sum_{\sigma \in S_k} \frac{1}{k!} \sum_{\tau \in S_k} h_k(x_{\tau(1)}, \ldots, x_{\tau(k)}) \qquad (38) \\
&= \frac{(n-k)!}{n!} \sum_{\boldsymbol{x} \in C(n,k;\boldsymbol{a})} \sum_{\tau \in S_k} h_k(x_{\tau(1)}, \ldots, x_{\tau(k)}) \\
&= \mathbb{E}\left[h_k(a_{i_1}, \ldots, a_{i_k})\right],
\end{aligned}
$$

where (38) follows because the summation over $\tau$ sums over the same set of summands. $\qquad \square$

# D   Sampling Distributions Depending On All Logits

In this section, we show that [3] and [16] do not meet Assumption 3.2 by giving a brief overview of how they work.

Their methods depend on the following problem structure. Given input space $\mathcal{X}$ and the input $\boldsymbol{x} \in \mathcal{X}$, logits of class $i$ take the form

$$\hat{y}_i = \langle f(\boldsymbol{x}), \boldsymbol{v}_i \rangle,$$

where $f : \mathcal{X} \to \mathbb{R}^d$ is the model, $d$ is the embedding dimension and $\boldsymbol{v}_i \in \mathbb{R}^d$ is the embedding of class $i$. In other words, the probability of class $i$ is given by

$$
\begin{aligned}
\text{Softmax}(\hat{y}_i, \hat{y}_1, \ldots, \hat{y}_{i-1}, \hat{y}_{i+1}, \ldots, \hat{y}_{n+1}) &= \frac{e^{\hat{y}_i}}{\sum_{j=1}^{n+1} e^{\hat{y}_j}} \\
&= \frac{e^{\langle f(\boldsymbol{x}), \boldsymbol{v}_i \rangle}}{\sum_{j=1}^{n+1} e^{\langle f(\boldsymbol{x}), \boldsymbol{v}_j \rangle}}.
\end{aligned}
$$

In keeping with (1), we let $n$ denote the number of negative classes, so there are a total of $n+1$ classes. They proposed a sampling distribution $p$ that is efficient to compute and has a structure similar to the softmax function. The probability $p(i)$ of sampling class $i$ is given by

$$p(i) = \frac{\langle \phi(f(\boldsymbol{x})), \phi(\boldsymbol{v}_i) \rangle}{\sum_{j=1}^{n+1} \langle \phi(f(\boldsymbol{x})), \phi(\boldsymbol{v}_j) \rangle}, \qquad (39)$$

where $\phi : \mathbb{R}^d \to \mathbb{R}^{d^2}$ is a feature map given by

$$\phi(\boldsymbol{v}) = (v_1 v_1, \ldots, v_1 v_d, v_2 v_1, \ldots, v_2 v_d, \ldots, v_d v_d).$$

Notably, the sampling distribution (39) does not satisfy Assumption 3.2 because (39) effectively involves computing the logits. Expanding out the terms of the inner product, we have

$$
\begin{aligned}
\langle \phi(f(\boldsymbol{x})), \phi(\boldsymbol{v}_i) \rangle &= \sum_{j=1}^{d} \sum_{k=1}^{d} f(\boldsymbol{x})_j f(\boldsymbol{x})_k v_{ij} v_{ik} \\
&= \sum_{j=1}^{d} f(\boldsymbol{x})_j v_{ij} \sum_{k=1}^{d} f(\boldsymbol{x})_k v_{ik} \\
&= \langle f(\boldsymbol{x}), \boldsymbol{v}_i \rangle \cdot \langle f(\boldsymbol{x}), \boldsymbol{v}_i \rangle \\
&= \hat{y}_i^2.
\end{aligned}
$$

# E  Proof of Corollary 3.3

*Proof.* The proof is by contradiction. Assume there is an unbiased estimator $h_k$ for $F$ with a sampling distribution $p$. That is,

$$
\Pr\left(i_1 = j_1, \ldots, i_k = j_k\right) = p(j_1, \ldots, j_k).
$$

The expectation is given by

$$
F(\boldsymbol{a}) = \mathbb{E}_p\left[h_k(a_{i_1}, \ldots, a_{i_k})\right] = \frac{1}{(n-k)!} \sum_{\sigma \in S_n} p(\sigma(1), \ldots, \sigma(k)) h_k(a_{\sigma(1)}, \ldots, a_{\sigma(k)}). \tag{40}
$$

Note that this is different from (12), in which we have $k$ choices from items first due to the uniform sampling. Therefore, here we have $S_n$ instead of $S_k$ in (12). Further, the summation over $\sigma$ has $n!$ summands but there are only

$$
\binom{n}{k} \cdot k! = \frac{n!}{(n-k)!}
$$

choices of indices. Therefore, the summation counts each choice of indices $(n-k)!$ times, which is corrected by the first factor $\frac{1}{(n-k)!}$.

We first observe that $F$ is symmetric in its arguments, i.e., for every permutation $\tau \in S_n$,

$$
F(\boldsymbol{a}) = \frac{1}{1 + \sum_{i=1}^{n} a_i} = \frac{1}{1 + \sum_{i=1}^{n} a_{\tau(i)}} = F(a_{\tau(1)}, \ldots, a_{\tau(n)}). \tag{41}
$$

By Assumption 3.2, $p$ is the same function regardless of the values of $\boldsymbol{a}$.[4] This property and (41) imply that, for every permutation $\tau \in S_n$,

$$
\begin{aligned}
F(\boldsymbol{a}) &= F(a_{\tau(1)}, \ldots, a_{\tau(n)}) \\
&= \frac{1}{(n-k)!} \sum_{\sigma \in S_n} p(\sigma(1), \ldots, \sigma(k)) h_k(a_{\sigma(\tau(1))}, \ldots, a_{\sigma(\tau(k))}).
\end{aligned} \tag{42}
$$

---

[4]Without Assumption 3.2, it is possible to have a different distribution $p_{\boldsymbol{a}}$ for each $\boldsymbol{a}$. See Appendix D for more details.

Therefore,

$$F(\boldsymbol{a}) = \frac{1}{n!} \sum_{\tau \in S_n} F(\boldsymbol{a})$$

$$= \frac{1}{n!} \sum_{\tau \in S_n} \frac{1}{(n-k)!} \sum_{\sigma \in S_n} p(\sigma(1), \ldots, \sigma(k)) \times h_k(a_{\sigma(\tau(1))}, \ldots, a_{\sigma(\tau(k))}) \qquad (43)$$

$$= \frac{1}{n!(n-k)!} \sum_{\sigma \in S_n} p(\sigma(1), \ldots, \sigma(k)) \times \sum_{\tau \in S_n} h_k(a_{\sigma(\tau(1))}, \ldots, a_{\sigma(\tau(k))})$$

$$= \frac{1}{n!(n-k)!} \sum_{\sigma \in S_n} p(\sigma(1), \ldots, \sigma(k)) \times \sum_{\tau \in S_n} h_k(a_{\tau(1)}, \ldots, a_{\tau(k)}) \qquad (44)$$

$$= \frac{1}{n!(n-k)!} (n-k)! \times \sum_{\tau \in S_n} h_k(a_{\tau(1)}, \ldots, a_{\tau(k)}), \qquad (45)$$

$$= \frac{1}{n!} \sum_{\tau \in S_n} h_k(a_{\tau(1)}, \ldots, a_{\tau(k)})$$

$$= \frac{(n-k)!}{n!} \sum_{x \in C(n,k;\boldsymbol{a})} \sum_{\sigma \in S_k} h_k(x_{\sigma(1)}, \ldots, x_{\sigma(k)}), \qquad (46)$$

where (43) follows from (42), and (44) follows because the summation over $\tau$ sums over the same set of summands. From (44) to (45), we use

$$\sum_{\sigma \in S_n} p(\sigma(1), \ldots, \sigma(k)) = (n-k)!$$

because, as we mentioned earlier, each element is counted $(n-k)!$ times. Finally, (46) is just (12), the expectation under the uniform distribution. This implies that $h_k$ is also an unbiased estimator under the uniform distribution. Such an $h_k$ does not exist by Theorem 3.1, leading to a contradiction. $\qquad \square$

## F   Impossibility of an Unbiased Log-Softmax Estimator

In this section, we prove that an unbiased and differentiable log-softmax estimator $\ell$ does not exist. We prove by contradiction. Assume that

$$\mathbb{E}\left[\ell(\hat{y}^+, \hat{y}_{i_1}^-, \ldots, \hat{y}_{i_k}^-)\right] = \log \operatorname{Softmax}(\hat{y}^+, \hat{y}_1^-, \ldots, \hat{y}_n^-).$$

Then

$$\frac{\partial}{\partial \hat{y}^+} \log \operatorname{Softmax}(\hat{y}^+, \hat{y}_1^-, \ldots, \hat{y}_n^-) = \frac{\partial}{\partial \hat{y}^+} \mathbb{E}\left[\ell(\hat{y}^+, \hat{y}_{i_1}^-, \ldots, \hat{y}_{i_k}^-)\right]$$

$$= \frac{\partial}{\partial \hat{y}^+} \frac{1}{n!} \sum_{\sigma \in S_n} \ell(\hat{y}^+, \hat{y}_{\sigma(1)}^-, \ldots, \hat{y}_{\sigma(k)}^-)$$

$$= \frac{1}{n!} \sum_{\sigma \in S_n} \frac{\partial}{\partial \hat{y}^+} \ell(\hat{y}^+, \hat{y}_{\sigma(1)}^-, \ldots, \hat{y}_{\sigma(k)}^-)$$

$$= \mathbb{E}\left[\frac{\partial}{\partial \hat{y}^+} \ell(\hat{y}^+, \hat{y}_{i_1}^-, \ldots, \hat{y}_{i_k}^-)\right]. \qquad (47)$$

So

$$\operatorname{Softmax}(\hat{y}^+, \hat{y}_1^-, \ldots, \hat{y}_n^-) = -\frac{\partial}{\partial \hat{y}^+} \log \operatorname{Softmax}(\hat{y}^+, \hat{y}_1^-, \ldots, \hat{y}_n^-) - 1$$

$$= \mathbb{E}\left[-\frac{\partial}{\partial \hat{y}^+} \ell(\hat{y}^+, \hat{y}_{i_1}^-, \ldots, \hat{y}_{i_k}^-) - 1\right], \qquad (48)$$

where the first equality follows from (2) and the second equality follows from (47). The expression

$$-\frac{\partial}{\partial \hat{y}^+} \ell(\hat{y}^+, \hat{y}_{i_1}^-, \ldots, \hat{y}_{i_k}^-) - 1$$

in (48) is therefore an unbiased estimator for softmax, which contradicts Theorem 3.1.

Note that the proof above assumes a differentiable estimator $\ell$, but the proof of Theorem 3.1 does not. Since we take the gradient of the loss estimator during training, we can reasonably assume that practical loss estimators are differentiable.

