# OpenReview forum: "Sampled Estimators For Softmax Must Be Biased"
_NeurIPS.cc/2025/Conference — NeurIPS 2025 poster_

### Official Review · Reviewer_Uj1A · 2025-06-27

**Clarity:** 4
**Significance:** 2
**Originality:** 3
**Rating:** 4
**Confidence:** 2

**Summary:**

This paper proves that an unbiased estimator based on sampled indices for full softmax must be biased for general cases.

**Questions:**

- Line 122. Positive classes are not defined. Moreover, what if the positive classes of other instances are identical to the current instance? Are they still so-called negative?
- Assumption 2.3 excludes adaptive sampling methods. The authors argue that this requires computing the full softmax. But is it possible to select sampling distribution after obtaining logits with a sampling approach rather than computing the full softmax?
- Is it possible to quantify the bias?
- How do existing techniques align with the paper's conclusion? Are there inspirations for new bias-reduction methods if we still adopt the biased estimator?
- The sampled estimator is biased for general cases and the paper proves it by crafting counter-examples. What are the cases / conditions under which the estimator is unbiased?

**Ethical Concerns:**

["NO or VERY MINOR ethics concerns only"]

**Final Justification:**

I recommend borderline acceptance because this paper provides important conclusion for exact unbiasedness of sampled softmax estimators, but not a stronger acceptance level because the conclusion does not lead to any more significant insights on future designs of better estimators. My concerns have been addressed by the authors though. But the content and key take-aways do not seem to put this work certainly above the acceptance borderline.

**Limitations:**

Yes.

**Paper Formatting Concerns:**

No.

**Quality:**

3

**Strengths And Weaknesses:**

**Strength**
- The results in the manuscript are mathematically sound with rigorous and detailed proof presented. The assumptions are mild, practical and reasonable.
- The conclusion is of interest to machine learning systems involving millions of possible classes to predict.

**Weaknesses**
- The results simply prevent investigations for finding unbiased estimators but do not provide any other deeper insights such as how to reduce the bias with the help of conclusions in the paper.

---

> ### Author Rebuttal · Authors · 2025-07-30
>
> > The results simply prevent investigations for finding unbiased estimators but do not provide any other deeper insights such as how to reduce the bias with the help of conclusions in the paper.
>
> While we agree that understanding and reducing bias is important, we chose to focus this paper specifically on the exact unbiased case as a first step.
>
> > Line 122. Positive classes are not defined.
>
> We should've written "target classes" instead of "positive classes". We thank the reviewer for pointing out our error.
>
> > Moreover, what if the positive classes of other instances are identical to the current instance? Are they still so-called negative?
>
> There are variants of in-batch sampling which do consider those classes as negative and there are variants which don't.
>
> > Assumption 2.3 excludes adaptive sampling methods. The authors argue that this requires computing the full softmax. But is it possible to select sampling distribution after obtaining logits with a sampling approach rather than computing the full softmax?
>
> It is possible and it may be a very fruitful future direction. We gave a brief dicussion in Appendix D on previous works that take this approach to reduce the bias. However, we are not aware of adaptive sampling methods that can eliminate the bias completely.
>
> > Is it possible to quantify the bias?
>
> To quantify the bias, we have to select a specific distribution of logits and estimator. Otherwise, any bias is possible. A trivial example is constant logits and using a constant function as estimator. An interesting future direction may be to empirically identify specific distributions and design estimators around them.
>
> > How do existing techniques align with the paper's conclusion? Are there inspirations for new bias-reduction methods if we still adopt the biased estimator?
>
> Existing techniques for softmax approximation are typically evaluated based on empirical performance, often without theoretical guarantees. Our result provides a formal justification for the use of biased estimators by showing that exact unbiasedness is unattainable, framing bias-reduction as a necessary tradeoff rather than a design flaw.
>
> As noted earlier, adaptive sampling methods are not covered by our impossibility result and may offer promising avenues for bias reduction. We see this as an exciting direction for future work.
>
> > The sampled estimator is biased for general cases and the paper proves it by crafting counter-examples. What are the cases / conditions under which the estimator is unbiased?
>
> Citing our response to reviewer 8WoT: There are trivial distributions (e.g., constant logits) under which an unbiased estimator could exist, but such cases are highly artificial and we are not aware of any that arise in practical settings.

---

> > ### Comment · Reviewer_Uj1A · 2025-08-05
> >
> > Thanks for your rebuttal. Most of my concerns have been addressed. The proof and conclusion in this manuscript are OK and should be helpful to the area of interest. However, since the applications and significance of simply disproving unbiasedness for sampled estimators of softmax may be limited, and an updated version of this manuscript with more content is beyond scope of the discussion, I will not improve my score for this paper.

---

### Official Review · Reviewer_8WoT · 2025-06-30

**Clarity:** 4
**Significance:** 2
**Originality:** 2
**Rating:** 3
**Confidence:** 2

**Summary:**

The paper provides a proof that no unbiased sampled estimator of the softmax function exists. The authors argue that this long suspected but never formally proven property could guide research into future softmax estimators.

The authors also give a thorough review of several related works which attempted to mitigate or avoid this bias.

The paper’s proof leverages a contradiction, showing that an estimator cannot exist which is unbiased in two cases: one where all logits are constant and one where all logits take on one of two values.

**Questions:**

Q1 What sorts of distributional assumptions on the logits would be required to construct an unbiased estimator for k << n?

Q2 Given a fixed k << n, can one get close to an unbiased estimator?

Q3 Does this property have implications for adaptive softmax estimators or softmax gradient estimators?

**Ethical Concerns:**

["NO or VERY MINOR ethics concerns only"]

**Final Justification:**

I still believe that the sole result of the paper lacks sufficient impact. The paper shows that if no restrictions are made on the distribution of logit vectors in a softmax, then an unbiased estimator cannot be created. The authors frame this lack of assumptions as a strength, but in reality it expands the family of allowed vectors to include adversarial examples which in turn makes the proof easier. Additionally, if we place no restrictions on the logit vectors, it seems unlikely one can expect to construct a useful estimator, much less an unbiased one.

This is why I strongly believe that in order to be published it is necessary for this paper to include further results bounding the bias of softmax estimators or provide a similar impossibility result for unbiased estimators given bounded logit distributions.

As a result, I have maintained my previous score.

**Limitations:**

yes

**Quality:**

2

**Strengths And Weaknesses:**

The provided proof is clear, formal, and relatively easy to follow. While the simplicity of the proof does not disqualify it from publication, it represents the sole output of the paper, meaning the bar for novelty and impact should be set high.

Unfortunately, the applications of the proved result appear relatively limited. No scenarios come to mind in which one would expect a softmax estimator to be unbiased across the entire space of logit vectors.

Though this analysis certainly helps to guide the conversation, it seems reasonable to expect further analysis from a publication of NeurIPS’ caliber (e.g., bounds on the bias based on k or distributional assumptions under which an unbiased estimator is possible).

---

> ### Author Rebuttal · Authors · 2025-07-30
>
> > Unfortunately, the applications of the proved result appear relatively limited. No scenarios come to mind in which one would expect a softmax estimator to be unbiased across the entire space of logit vectors.
>
> While the result may align with intuition in hindsight, to our knowledge, no formal proof previously existed. Moreover, unlike softmax, there do exist unbiased estimators for other commonly used objectives (e.g., binary cross-entropy), which makes the softmax case non-obvious and worth settling formally.
>
> > Though this analysis certainly helps to guide the conversation, it seems reasonable to expect further analysis from a publication of NeurIPS’ caliber (e.g., bounds on the bias based on k or distributional assumptions under which an unbiased estimator is possible).
>
> We agree that further analysis (e.g., bias bounds or conditions for approximability) would be valuable, but we chose to focus this work on conclusively resolving the exact unbiasedness question, which we view as a fundamental theoretical contribution.
>
> > What sorts of distributional assumptions on the logits would be required to construct an unbiased estimator for k << n?
>
> There are trivial distributions (e.g., constant logits) under which an unbiased estimator could exist, but such cases are highly artificial and we are not aware of any that arise in practical settings.
>
> > Given a fixed k << n, can one get close to an unbiased estimator?
>
> We did not study approximation quality or bias-variance tradeoffs, as our focus in this work was on establishing the impossibility of exact unbiasedness. We agree this is an important direction for future research.
>
> > Does this property have implications for adaptive softmax estimators or softmax gradient estimators?
>
> We're not sure what is meant by "adaptive softmax estimators," but if this refers to methods that adjust the sampling process based on observed data, then we have a discussion in Appendix D.

---

> > ### Comment · Reviewer_8WoT · 2025-08-07
> >
> > Thank you for the response.
> >
> > I ask about the adaptive methods and distributional assumptions on the logits because existing work ("Adaptive Sampling for Efficient Softmax Approximation" [Baharav et. al]) provided PAC-guarantees on the softmax assuming sub-Gaussianity bounds on the logits. As a result, it's not clear to me that unbiased estimators can't exist for non-trivial distributions. Since the paper only removes the possibility of producing unbiased estimators that sample wor from an unrestrained vector of logits, my concerns about limited impact still stand.
> >
> > As a result, I will maintain my previous score.

---

> > > ### Author Response · Authors · 2025-08-09
> > >
> > > We thank the reviewer for highlighting this related work. We believe our paper provides strong motivation for seeking approximations as discussed in [Baharav et al.]. Without first establishing that unbiased sampled estimators do not exist, it is more difficult to justify the need for such approximations.
> > >
> > > We also note that [Baharav et al.] addresses a different setting: their method focuses on approximating the probability of only the top few classes. This approach is unsuitable for training, where the softmax over all classes must be computed. Moreover, as the reviewer mentioned, [Baharav et al.] relies on specific assumptions about the logits. In contrast, our theorem is derived under minimal assumptions, making it broadly applicable to anyone working on estimating the softmax function.

---

### Official Review · Reviewer_gnrp · 2025-06-30

**Clarity:** 3
**Significance:** 2
**Originality:** 3
**Rating:** 4
**Confidence:** 3

**Summary:**

The authors consider the problem of approximating the softmax using sampling of its arguments. They prove theoretically that any sampling-based estimator that samples a fixed number of logits (and that is not allowed to depend on the logits) must necessarily be biased.

**Questions:**

- Does the analysis shed any light on the difficulty of *approximating* the softmax?  This would be useful for estimators that navigate a bias-variance tradeoff.
- The analysis also doesn’t seem to rule out estimators that use a stochastic number of samples (e.g., based on a stopping rule), correct?  If it does, it would be great to point it out.

**Ethical Concerns:**

["NO or VERY MINOR ethics concerns only"]

**Final Justification:**

I'd like to thank the authors for the clarification.  While the considered sampling scheme (and hence the implied hardness result) indeed is relevant for approximating the gradient as well (as discussed in Appendix A), the problem of approximating the log softmax itself (i.e., for computing the loss and not its gradient) is still relevant and not considered.  Nevertheless, while I still feel that the paper is somewhat limited, in my opinion it makes a relevant contribution to the literature and I have increased my score.

**Limitations:**

Not explicitly discussed; some of the points mentioned above could be mentioned.

**Quality:**

3

**Strengths And Weaknesses:**

Strengths:
- The negative result provided by the paper is interesting, and supports / encourages the development of efficient biased estimators

Weaknesses:
- While the construction provided rules out unbiased estimates for the softmax, I don’t think it technically rules out unbiased estimates of, e.g., the log softmax (which is typically used in loss computation), which would be sufficient to obtain unbiased stochastic gradient estimates.  It would be great if the authors could comment on this.
- Moreover, the theoretical analysis rules out *exactly* unbiased estimators, leaving open important questions of approximability (see below).

---

> ### Author Rebuttal · Authors · 2025-07-30
>
> > While the construction provided rules out unbiased estimates for the softmax, I don’t think it technically rules out unbiased estimates of, e.g., the log softmax (which is typically used in loss computation), which would be sufficient to obtain unbiased stochastic gradient estimates. It would be great if the authors could comment on this.
>
> We would like to clarify that we specifically addressed the gradient of log softmax in Appendix A. We showed that estimating the gradient of log softmax is precisely the same problem as estimating softmax. Therefore, we did rule out unbiased estimates for the log softmax as a loss.
>
> > Does the analysis shed any light on the difficulty of approximating the softmax? This would be useful for estimators that navigate a bias-variance tradeoff.
>
> We agree that understanding approximability and bias-variance tradeoffs is important, and we are interested in pursuing this in future work. However, our focus in this paper was to resolve the question of exact unbiasedness, which remained open.
>
> > The analysis also doesn’t seem to rule out estimators that use a stochastic number of samples (e.g., based on a stopping rule), correct? If it does, it would be great to point it out.
>
> We're not sure what was meant by estimators using a stochastic number of samples. If the suggestion refers to iterative schemes that incrementally sample more logits, then we think this is an interesting direction. Our analysis does not address such schemes, and we appreciate the suggestion as a potential future direction.

---

> > ### Comment · Reviewer_gnrp · 2025-08-02
> >
> > I'd like to thank the authors for the clarification. While the considered sampling scheme (and hence the implied impossibility result for unbiased sampling) indeed is relevant for approximating the gradient as well (as discussed in Appendix A), the problem of approximating the log softmax itself (i.e., for computing the loss and not its gradient) is still relevant and not considered.

---

> > > ### Comment · Area_Chair_kTEx · 2025-08-05
> > >
> > > Dear reviewer gnrp and authors,
> > >
> > > Thank you for the discussion on the difference (or equivalence) between estimating the softmax vs. estimating the gradient.
> > >
> > > I have a followup question:
> > >
> > > In Blanc and Rendle [2] Theorem 2.1 they already showed that
> > > The gradient of sample softmax is an unbiased estimator of the full softmax gradient iff qi = pi ∝exp(oi).
> > >
> > > In other words, the sampled distribution must be softmax itself in order for the estimate to be unbiased.
> > >
> > > Isn't the above a stronger result?
> > >
> > > AC

---

> > > > ### Author Response · Authors · 2025-08-06
> > > >
> > > > > In Blanc and Rendle [2] Theorem 2.1 they already showed that The gradient of sample softmax is an unbiased estimator of the full softmax gradient iff qi = pi ∝exp(oi).
> > > > > In other words, the sampled distribution must be softmax itself in order for the estimate to be unbiased.
> > > > > Isn't the above a stronger result?
> > > >
> > > > Blanc and Rendle [2] Theorem 2.1 states that if you use the sampled softmax as an estimator, which is the specific function
> > > > $$
> > > > \\begin{align*}
> > > >     \\text{SampledSoftmax}(\\hat{y}^+, \\hat{y}^-\_{i\_1}, \\ldots, \\hat{y}^-\_{i\_k})
> > > >     &= \\frac{e^{\\hat{y}^+}}{e^{\\hat{y}^+} + \\sum\_{j=1}^{k} w(i\_j) e^{\\hat{y}^-\_{i\_j}}},
> > > > \\end{align*}
> > > > $$
> > > > then it is unbiased iff $q\_i = p\_i \\propto \\exp(o\_i)$. However, this leaves open the question whether it is possible to add extra terms to the function that results in an unbiased estimator.
> > > >
> > > > Our paper presents a significantly more general result. Theorem 3.1 states that *any* estimator $s: \\mathbb{R}^{k+1} \\to \\mathbb{R}$ must be biased. This includes the sampled softmax as a special case. Not only does Theorem 3.1 rule out the possibility of adding terms to the sampled softmax to remove the bias, Theorem 3.1 also rules out entirely different unbiased estimators. Theorem 3.1 assumes nothing about $s$: $s$ can be continuous or discontinous, $s$ can be computable or uncomputable-*every* estimator $s$ is biased.

---

> > > ### Author Response · Authors · 2025-08-06
> > >
> > > > I'd like to thank the authors for the clarification. While the considered sampling scheme (and hence the implied impossibility result for unbiased sampling) indeed is relevant for approximating the gradient as well (as discussed in Appendix A), the problem of approximating the log softmax itself (i.e., for computing the loss and not its gradient) is still relevant and not considered.
> > >
> > > We argue that estimating log softmax accurately is not crucial in machine learning. During training, the value of interest is its gradient because the important operation is updating model weights with the gradient. During prediction, the value of interest is softmax because that represents the probability of each class. In both scenarios, the precise value of log softmax is not used.
> > >
> > > However, we can also prove that log softmax must be biased and thank the reviewer for bringing this up. The proof is given as follows.
> > >
> > > We prove by contradiction. Assume that there exists a differentiable unbiased estimator $l$ for log softmax
> > > $$
> > > \\begin{align*}
> > >     \\mathbb{E} \\left[
> > >             l(\\hat{y}^+, \\hat{y}^-\_{i\_1}, \\ldots, \\hat{y}^-\_{i\_k})
> > >         \\right]
> > >     = \\log \\text{Softmax}(\\hat{y}^+, \\hat{y}^-\_{1}, \\ldots, \\hat{y}^-\_{n}).
> > > \\end{align*}
> > > $$
> > > Then
> > > $$
> > > \\begin{align*}
> > >     \\frac{\\partial}{\\partial \\hat{y}^+} \\log \\text{Softmax}(\\hat{y}^+, \\hat{y}^-\_{1}, \\ldots, \\hat{y}^-\_{n})
> > >     &= \\frac{\\partial}{\\partial \\hat{y}^+} \\mathbb{E} \\left[
> > >             l(\\hat{y}^+, \\hat{y}^-\_{i\_1}, \\ldots, \\hat{y}^-\_{i\_k})
> > >         \\right] \\\\
> > >     &= \\frac{\\partial}{\\partial \\hat{y}^+} \\frac{1}{n!} \\sum\_{\\sigma \\in S\_n} l(\\hat{y}^+, \\hat{y}^-\_{\\sigma(1)}, \\ldots, \\hat{y}^-\_{\\sigma(k)}) \\\\
> > >     &= \\frac{1}{n!} \\sum\_{\\sigma \\in S\_n} \\frac{\\partial}{\\partial \\hat{y}^+} l(\\hat{y}^+, \\hat{y}^-\_{\\sigma(1)}, \\ldots, \\hat{y}^-\_{\\sigma(k)}) \\\\
> > >     &= \\mathbb{E} \\left[
> > >             \\frac{\\partial}{\\partial \\hat{y}^+} l(\\hat{y}^+, \\hat{y}^-\_{i\_1}, \\ldots, \\hat{y}^-\_{i\_k})
> > >         \\right].
> > > \\end{align*}
> > > $$
> > > So
> > > $$
> > > \\begin{align*}
> > >     \\text{Softmax}(\\hat{y}^+, \\hat{y}^-\_{1}, \\ldots, \\hat{y}^-\_{n})
> > >     &= -\\frac{\\partial}{\\partial \\hat{y}^+} \\log \\text{Softmax}(\\hat{y}^+, \\hat{y}^-\_{1}, \\ldots, \\hat{y}^-\_{n}) - 1 \\\\
> > >     &= \\mathbb{E} \\left[
> > >             -\\frac{\\partial}{\\partial \\hat{y}^+} l(\\hat{y}^+, \\hat{y}^-\_{i\_1}, \\ldots, \\hat{y}^-\_{i\_k}) - 1
> > >         \\right],
> > > \\end{align*}
> > > $$
> > > where the first equality is derived in Appendix A from the definition of softmax and the second equality is applying the result from above. Therefore,
> > > $$
> > > \\begin{align*}
> > >     -\\frac{\\partial}{\\partial \\hat{y}^+} l(\\hat{y}^+, \\hat{y}^-\_{i\_1}, \\ldots, \\hat{y}^-\_{i\_k}) - 1
> > > \\end{align*}
> > > $$
> > > is an unbiased estimator for softmax, which contradicts Theorem 3.1.
> > >
> > > Note that the proof above assumes a differentiable estimator $l$, but the proof of Theorem 3.1 does not. Since we take the gradient of the loss estimator during training, we can reasonably assume that practical loss estimators are differentiable.

---

### Official Review · Reviewer_Y1Xs · 2025-07-03

**Clarity:** 3
**Significance:** 2
**Originality:** 2
**Rating:** 4
**Confidence:** 4

**Summary:**

In this work, the authors address a problem related to the use of the softmax function over a large number of classes. To handle the computational cost of the full softmax, practitioners often use a sampled softmax, which is known to be a biased estimator. This paper proves that any estimator that relies on a sampled subset of classes must be biased, under the assumption that the sampling distribution is independent of the logit values. The main contribution is a formal impossibility proof that confirms a widely held belief in the field, suggesting that research should continue to focus on bias reduction rather than elimination.

**Questions:**

1. The proof relies on constructing specific configurations of the logit values (e.g., all equal, or split into two groups). This is a valid and powerful proof technique. Have the authors considered alternative proof strategies, perhaps from information theory or functional analysis, that might offer a different perspective on why this unbiased estimation is impossible?

2. The problem definition and proof are based on sampling k indices from n without replacement. Does the impossibility result also hold for sampling with replacement? It seems intuitively that it should, as this would be a less informative sample.

**Ethical Concerns:**

["NO or VERY MINOR ethics concerns only"]

**Final Justification:**

This is a solid paper that rigorously proves something important: any estimator using a sampled subset of classes for a softmax function will be biased. This formalizes a belief many in the field already held, which is a valuable contribution.

The main drawback is that it doesn't really open up new research paths; it mostly just confirms that the current focus on reducing bias is the correct approach. Because the impact is more foundational than groundbreaking, the paper is seen as technically sound but a bit incremental. The good (a rigorous proof) just outweighs the bad (limited novelty).

**Limitations:**

Yes, the authors have adequately addressed the limitations. They are very clear that their work is a theoretical impossibility proof and does not provide a new algorithm.

**Paper Formatting Concerns:**

None.

**Quality:**

3

**Strengths And Weaknesses:**

#### Strengths

1. The paper provides a solid and rigorous proof for a widely-held, but previously unproven, belief in the machine learning community. Formalizing this impossibility result is a valuable contribution that helps solidify the theoretical foundations of large-scale classification.

2. The paper is well-written, and the proof is presented clearly. The logical flow is easy to follow, and the authors do a good job of setting up the problem and their assumptions. The technical quality of the mathematical argument is high.

#### Weaknesses

1. While the result is a useful formalization, it proves something that was already strongly suspected and implicitly assumed by most researchers in the area. Therefore, the work confirms the existing research direction (bias reduction) rather than opening up new ones. Its overall impact might be limited.

2. The work is entirely theoretical. It does not offer new insights into the nature or magnitude of the bias for different estimators, nor does it provide any guidance for designing better-biased estimators. The lack of any empirical validation or practical takeaways makes the contribution feel incremental.

---

> ### Author Rebuttal · Authors · 2025-07-30
>
> > While the result is a useful formalization, it proves something that was already strongly suspected and implicitly assumed by most researchers in the area. Therefore, the work confirms the existing research direction (bias reduction) rather than opening up new ones. Its overall impact might be limited.
> > The work is entirely theoretical. It does not offer new insights into the nature or magnitude of the bias for different estimators, nor does it provide any guidance for designing better-biased estimators. The lack of any empirical validation or practical takeaways makes the contribution feel incremental.
>
> We wholeheartedly agree with the reviewer's observation that our result formalizes an assumption widely held in the community. It is precisely the widespread relevance of our research question that makes our conclusion an important addition to the published literature. While the conclusion may have been strongly suspected, our goal was to rigorously prove it and settle the matter definitively. We believe resolving this foundational question helps solidify the basis for ongoing work in bias reduction.
>
> > The proof relies on constructing specific configurations of the logit values (e.g., all equal, or split into two groups). This is a valid and powerful proof technique. Have the authors considered alternative proof strategies, perhaps from information theory or functional analysis, that might offer a different perspective on why this unbiased estimation is impossible?
>
> Regarding alternative proof strategies, we did attempt several directions during the course of our work. However, the construction-based approach was the only one that led to a complete proof. We agree that alternative perspectives could offer valuable insight and worth exploring further in future work.
>
> > The problem definition and proof are based on sampling k indices from n without replacement. Does the impossibility result also hold for sampling with replacement? It seems intuitively that it should, as this would be a less informative sample.
>
> Our premise of sampling without replacement is based on the most common practice in real-world applications. We agree that extending the result to sampling with replacement is a natural next step and thank the reviewer for the suggestion.

---

> > ### Comment · Reviewer_Y1Xs · 2025-08-04
> >
> > Thank you for your detailed response to our initial review. Your clarifications are helpful. On the topic of sampling with replacement, we appreciate your reasoning for focusing on the more common "without replacement" scenario. Based on your deep work on the proof, what is your expert intuition on the main challenge in extending your result to the "with replacement" case? Would the combinatorial arguments and the derivation of the coefficients c(m,l) need a fundamental rethinking, or do you believe a similar contradiction could be constructed with some modifications?

---

> > > ### Author Response · Authors · 2025-08-06
> > >
> > > > Thank you for your detailed response to our initial review. Your clarifications are helpful. On the topic of sampling with replacement, we appreciate your reasoning for focusing on the more common "without replacement" scenario. Based on your deep work on the proof, what is your expert intuition on the main challenge in extending your result to the "with replacement" case? Would the combinatorial arguments and the derivation of the coefficients c(m,l) need a fundamental rethinking, or do you believe a similar contradiction could be constructed with some modifications?
> > >
> > > In the case of sampling without replacement, the proof strategy is to find specific $\boldsymbol{a}$ that "forces" the estimator to take on specific values for specific arguments. The difficulty for sampling with replacement lies in (17), where instead of there being two possible combinations of $b_1$ and $b_2$, there will be $(k + 1)$ possible combinations. Therefore, instead of discovering what $f_k(b_1,\ldots,b_1,b_2)$ must be, we will have an equation with $k$ unknown function values.
> > >
> > > While we have not finished the proof, here is a proof sketch. Similarly consider $\boldsymbol{a}$ of the form
> > > $$
> > > \begin{align*}
> > >     \boldsymbol{a} = (b_1, \ldots, b_1, \underbrace{b_2, \ldots, b_2}_{m})
> > > \end{align*}
> > > $$
> > > for $1 \le m \le k + 1$. Then we will have a linear system of $k$ unknowns and $(k + 1)$ equations. Showing that the system does not have a solution finishes the proof.

---

### Decision · Program_Chairs · 2025-09-17

**Decision:**

Accept (poster)

**Comment:**

This paper addresses a fundamental problem in large-scale classification: efficient approximation of the full softmax.
It is well known that sampled softmax is biased. But it remains to be an open problem whether one can design an unbiased estimator of the full softmax. This work shows that ANY estimator that only accesses a sampled subset of classes is biased. This is a good validation on the many past works and therefore a very meaningful contribution to the community.

There has been a healthy amount of discussions during the rebuttal period.
After the rebuttal all reviewers recognized the technical soundness of the work.
The remaining concerns shared by reviewers are that the result is "not surprising" and therefore it does not "open up new research path". In particular the only reviewer who rated the paper below 4 states that the work "lacks sufficient impact".

The machine learning community has long operated under the assumption that sampled softmax estimators are inherently biased, leading to a significant body of work on bias reduction. This paper provides the foundational result that this assumption is correct and that the pursuit of bias reduction is not just a practical choice but a theoretical necessity. Therefore the AC recommends that the paper to be accepted.

In the final version, in addition to incorporating the reviewers feedback, please
* add key proof ideas and sketch in the main text.
* add the equivalence between estimating softmax vs. estimating the gradient in the main text.